# Effects of Different Hamstring Eccentric Exercise Programs on Preventing Lower Extremity Injuries: A Systematic Review and Meta-Analysis

**DOI:** 10.3390/ijerph20032057

**Published:** 2023-01-23

**Authors:** Chenxi Hu, Zhikun Du, Mei Tao, Yafeng Song

**Affiliations:** Department of Chinese Academy of Sport and Health, Beijing Sport University, Beijing 100084, China

**Keywords:** hamstring, eccentric training, injuries, prevention, training program

## Abstract

This systematic review and meta-analysis aims to investigate the effects and differences of various hamstring eccentric training protocols for the prevention of lower limb injuries, and we further propose a more refined hamstring eccentric training protocol for the prevention of lower limb injuries. A literature search for the effects of hamstring eccentric training on lower extremity sports injuries was conducted using the PubMed, Web of Science, and EMBASE databases, and the literature was searched covering the period from the date of the database’s creation to 20 August 2022. A meta-analysis of the included literature was performed using R.4.21 for lower extremity injuries, injuries in various parts of the lower extremity, and subgroup analysis for exercise frequency, exercise cycle, and exercise population. A total of 23 randomized controlled trial (RCT) studies were found to be included in the meta-analysis, and 15 of these trials, totaling 14,721 patients, were determined to be included in the overall lower extremity injury prevention effect. The analysis showed that the implementation of a hamstring eccentric training program reduced lower extremity injuries by 28%, and it resulted in a 46% decrease in hamstring injury rate and a 34% decrease in knee injury rate. The subgroup analysis revealed that the frequency of exercise was most significant in the twice-a-week exercise group, that the exercise program was most effective in preventing injuries in the 21–30-week exercise period, and that the program was most effective in preventing injuries in elite athletes and amateur adult athletic populations, compared with adolescents.

## 1. Introduction

Sports have a significant contribution to physical health, but they also carry certain risks [1,2]. Epidemiological surveys have shown that the risk of injury is 1.5 to 2.0 times higher in individuals who regularly participate in various sports [3]. In sports such as basketball, soccer, and rugby, lower extremity injuries are common sports injuries gained during their training and competitions [4,5]. Sports injuries have imposed a significant economic burden on athletes and sports groups [6,7]. Sports injuries are most prevalent in adolescents and also have a significant impact on adults [8]. An Australian study estimated the direct cost of sports-related injuries at $265 million over seven years [3]. Lower extremity injuries account for more than 60% of the total sports injury burden, with 60% of these being ankle and knee injuries [9]. Hamstring strains account for 1/6 of injuries in the Australian Football League, with injured players missing almost the entire season. The treatment of sports injuries requires a long period of time, while the injury can affect the athletic ability and performance of the athlete [10]. A sports injury can cause an athlete to stop training or performing, and it is likely to affect the athlete’s athletic career [11].

Therefore, targeted injury prevention training is a very effective intervention in sports training [12,13,14,15]. In recent years, more and more athletes or teams are adopting injury prevention programs, which is an essential part of their warm-up or strength training [16,17]. The aim is to effectively reduce the incidence of sports injuries. “FIFA11+” is one of the most effective prevention programs which consists of three parts: basic running (Part 1); three levels of difficulty of six exercises aiming to increase muscular strength (core and lower limbs), balance, muscle control (plyometrics), and core stability (Part 2); and running (e.g., straight-line running or cutting activities) (Part 3). A related cluster randomized controlled trial (RCT) study, in which all 20 teams comprising 414 players aged 14–19 years in the English Premier League had implemented the FIFA11+ warm-up exercises in the intervention group and showed a 41% reduction in the overall injury probability in the intervention group, as compared with the control group [18]. A similar study showed a significant reduction in severe soccer sports injuries using an injury protection program that included hamstring centrifugation (FIFA11+) [19]. The intervention group had a 77.1% reduction in severe hamstring injuries, and multiple RCTs have demonstrated the feasibility of sports injury protection programs [20,21,22].

The hamstrings are muscles in the posterior thigh—consisting mainly of the semitendinosus, semimembranosus, and biceps femoris. The hamstrings begin at the sciatic tuberosity and end at the tibia, spanning the hip and knee joints. The main function of these muscles is to flex the knee and extend the hip, and another important role is to maintain the stability of the knee joint [23,24]. Significant disability, activity restrictions, and participation restrictions, including time away from competitive sports, may follow a hamstring injury. A hamstring injury may be connected to professional sports with substantial financial expenses. Another significant problem is the high rate of reinjuries [25]. Prospective studies have highlighted modifiable injury risk factors, such as eccentric knee flexor weakness and muscle architectural features (such as biceps femoris fascicle length), as a target for hamstring injury prevention strategies [26,27]. The Federation Internationale de Football Association (FIFA) Medical Assessment and Research Centre (F-MARC) has developed the “FIFA 11” and “FIFA 11+” prevention programs [28]. These programs consist of several exercises that take approximately 15 min before each workout. These include self-weighted hamstring eccentric training, for example, the Nordic hamstring exercise (NHE), with one set with eight repetitions [29]. Neuromuscular training is a comprehensive training method that includes strength, speed, agility, balance, and other athletic abilities [20]. Recent neuromuscular training and injury prevention training also use hamstring eccentric training as part of strength training or warm-up activation, with eccentric training using resistance or self-weight training forms to improve the eccentric muscle strength of the hamstrings to reduce the risk of injury [30,31]. A meta-analysis by AI Attar et al. showed that the implementation of the F-MARC protection program significantly reduced the overall risk of sports injuries in soccer [28]. A recent study showed a 49% reduction in the risk of hamstring injury with the addition of hamstring eccentric training during exercise [32]. However, most of the previous related research studies were limited to soccer, and a considerable number of RCTs were not included; also, no reviews or meta-analyses focused on integrating and comparing exercise protection measures in hamstring eccentric training programs, and they lacked a categorical study of the participating population, exercise levels, and training period [33,34,35]. The present review and meta-analysis will broaden the scope of the previous studies by focusing on the effectiveness of different eccentric training programs for injury prevention and the prevention of injuries in different locations of the lower extremity.

## 2. Materials and Methods

The review has been registered in the PROSPRERO database (CRD42022352757). This systematic review and meta-analysis was conducted using advance reporting from the PRISMA guidelines [36].

### 2.1. Eligibility Criteria

#### 2.1.1. Inclusion Criteria

The RCT studies selected for inclusion had to be written in the English language, and the selected literature needed to meet the following PICOS criteria:

(1) Population: All people of both sexes being involved in sports, including professional and semi-professional players, sports enthusiasts, and students in high school and college sports teams.

(2) Intervention: Warm-up programs, including hamstring eccentric training and injury prevention programs: FIFA, FIFA11+, FIFA11+kids, F-MARC protection program, NHE exercises, or integrated neuromuscular training.

(3) Control group: A conventional warm-up program or training program was adopted in the study, and the articles had to include the training method and training duration of the control group.

(4) Outcome: The studies needed to report the incidence of lower extremity injuries, including the number of lower extremity injuries, such as hip injuries, hamstring injuries, knee injuries, ankle injuries, or the number of injuries sustained during 1000 h of training or competition.

(5) Study type: Clinical trials or RCTs.

#### 2.1.2. Exclusion Criteria

Following the PICOS principles, the exclusion criteria considered were as follows:

(1) Population: People with no exercise capacity or with major diseases.

(2) Intervention: Hamstring eccentric training was not employed in the intervention or was performed after surgery.

(3) Control group: The control group had no intervention or used the same measures as the intervention group.

(4) Outcome: No reported number of lower extremity injuries or injury rates.

(5) Study type: Single-arm study, systematic reviews with meta-analysis, conference reports, case studies, and animal experiments.

### 2.2. Information Sources

The studies were retrieved both from electronic databases and comprehensive searches of reference lists of included studies. The search—covering a period between the creation of a database and 20 August 2022—included the following databases: PubMed, Web of Science, and EMBASE.

### 2.3. Search Strategy

An advanced search was performed with subject terms and free words, and the search terms were: “eccentric exercise, NHE, Nordic hamstring exercise, Russian leg curl, kneeling Russian hamstring curl, natural hamstring curl, and bodyweight and bodyweight, natural hamstring curl, bodyweight hamstring curl, FIFA11+, FIFA11, F-MARC, The 11+, warm up program, injury prevention program, neuromuscular training, lower limb injuries, injury, sport injuries, athletic injuries.” The search period was from the beginning of the database creation until 20 August 2022. The search lines have been provided in the Appendix A.

### 2.4. Selection Process

Document search records were managed using the EndNote (version X9) document management software. Based on the established inclusion and exclusion criteria, two reviewers (C.H. and Z.D.) independently reviewed the contents of the titles and abstracts and read the full texts of the studies that met the criteria for final inclusion. After completing the screening, the screening results of these two reviewers were compared, and, if there was a dispute, it was resolved through discussion with a third reviewer (M.T.).

### 2.5. Data Collection Process

Data for the articles were extracted independently by two reviewers (C.H. and Z.D.) in Microsoft Excel (version 2015). Disputes were resolved through consultation between the two reviewers, and, when they could not be resolved at their level, a third reviewer (M.T.) adjudicated. In cases where the data were incomplete, the author/s were contacted via email to obtain the complete data.

### 2.6. Data Items

The data to be extracted from the studies included: (1) basic information about the study, such as the name of the author and the year of publication; (2) population characteristics, such as the sport level, sample size, age, sex, etc.; (3) intervention measures, intervention period, and the intervention frequency; and (4) subject lower extremity injuries and the number and injury rate of each location injury. If a study reported results for different training periods, we selected the results for the longest intervention period. The primary outcome was lower extremity injuries. The secondary outcomes were hip, hamstring, knee, and ankle injuries. The number of injuries was defined as the number of injuries that occurred during a game or a training session.

### 2.7. Study Risk of Bias Assessment

Two investigators (C.H. and Z.D.) independently assessed the risk and quality of bias according to the Cochrane risk of bias tool (ROB) [37], which classified bias into six categories: selectivity bias, implementation bias, measurement bias, follow-up bias, and other bias. Each article was evaluated with “low risk,” “high risk,” and “unclear risk,” and disagreements, whenever encountered, were resolved by a third investigator.

### 2.8. Effect Measures

Lower extremity injury is a dichotomous variable, and this paper uses the risk ratio (RR), which is the incidence of injury in the intervention group vis-à-vis incidence of injury in the control group with 95% confidence interval as the combined effect size. If RR = 1, it implies that the intervention group is different from the control group; if RR > 1, it indicates that the intervention is harmful to the subjects; if RR < 1, it indicates that the intervention has a positive effect on the prevention of lower limb injuries; and the lower the RR value, the greater is the effectiveness of the intervention.

### 2.9. Synthesis Methods

The meta-analysis was completed following the Cochrane approach. The results from the RCTs were incorporated into this study for meta-analyses to guarantee homogeneity of the study design. The meta-analysis was performed using the R statistical software (‘R’ statistical environment V.4.2.1, www.r-project.org (accessed on 30 August 2022 )). People of varied ages and levels of exercise were included in this meta-analysis. We selected the random-effects model considering that the dose and methodology of the interventions varied. Heterogeneity in the effect of study-constructed interventions was tested by I^2^ in the Cochrane guidelines: if I^2^ < 50%, we used the fixed-effects model, and if I^2^ ≥ 50%, we used the random-effects model. The subgroup analysis was performed to find sources of heterogeneity. Publication bias has been illustrated with a funnel plot.

### 2.10. Publication Bias

Twenty-three papers were included in the study, and the publication bias was checked using Peters’ test and funnel plots were generated.

### 2.11. Assessment of Certainty of Evidence

The certainty of evidence was rated based on the GRADE recommended guidelines [38], according to which the level of evidence is reduced when there are serious concerns with respect to risk of bias, imprecision, inconsistency, indirectness, publication bias, and the level of risk is upgraded when there is a large magnitude of effect, dose response, or no plausible confounding. A summary of findings table was generated using the GRADE profiler software (version 3.6; Cochrane IMS) for the outcomes. The outcome indicators included lower extremity injury, hamstring injury, hip and groin injury, knee injury, and ankle injury. The quality of evidence across each outcome was eventually classified into four levels, namely, high, moderate, low, and very low. The results of the GRADE system are shown in Appendix A.

## 3. Results

### 3.1. Study Selection

The results of the search are shown in Figure 1. A total of 2478 papers were retrieved by searching the PubMed, Web of Science, and EMBASE databases; 48 papers were obtained after removing duplicates, animal experiments, non-randomized controlled experiments, and experimental papers that did not meet the inclusion criteria. After reading the full text of the literature of these 48 papers, 15 papers were retrieved in the end. By reading the meta-analysis of other researchers, 8 additional papers were included after a three-person discussion, and finally 23 RCT studies were included.

### 3.2. Study Characteristics

A total of 23 RCT studies were included in the study. Twenty studies in this literature were in the sport of soccer, one included the sport of soccer with lacrosse, one handball sport, and one study on the sport of basketball. Four studies were conducted in female athletic groups, 16 studies were conducted in male athletic groups, two studies were conducted with both male and female participants, and two studies did not report the gender of the participants.

Out of the twenty-three studies on outcome measures, seven reported on the number of hamstring injuries, three on the number of lower extremity injuries, one on anterior cruciate ligament (ACL) injuries, and twelve on systemic injuries. Exercise interventions included a comprehensive warm-up with exercises such as Nordic jerk, the inclusion of hamstring eccentric training in daily training, and exercises such as FIFA11+ and FIFA11+kids. The periodicity of the intervention ranged from 10 to 52 weeks and the frequency of the intervention ranged from one to five times a week. The basic characteristics of the included studies (*n* = 23) are shown in Appendix A.

### 3.3. Risk of Bias in Studies

The risk of bias assessment is summarized in Figure 2. High risk of bias and detection bias were identified, with moderate selection bias. Attrition and reporting bias were low for the included studies. Individual assessments of high, low, or unknown risk of bias for each study are shown in Figure 2.

### 3.4. Publication Bias Results

An examination of the funnel plot shows slight asymmetry, suggesting that bias may not be present (Figure 3). The Peters’ test confirmed symmetry (intercept = −0.2216; SE = 0.0905; *p* = 0.28).

### 3.5. Effect of Hamstring Eccentric Exercise on Preventing Lower Extremity Injuries

#### 3.5.1. Overall Effect Test

Twenty-three randomized controlled trials, with a total of 18,215 subjects, were included in the literature for analysis. A total of 15,049 lower extremity injury subjects from 15 publications were included in the overall effect test for the overall effect analysis (Figure 4) [4,5,19,22,27,39,40,41,42,43,44,45,46,47]. The results suggested an effect size of RR = 0.72 (95% CI: 0.71–0.80; Z = −9.06; *p* < 0.01), indicating that a program of eccentric training of the hamstring muscle is effective in preventing the risk of lower extremity injury in athletic populations, and a test of heterogeneity was also performed for the included studies (I^2^ = 83%; *p* < 0.05) indicating a high degree of heterogeneity between studies. Therefore, the authors have used the random-effects model in the meta-analysis.

#### 3.5.2. Injury in Different Body Regions

Secondary analyses of the effect of hamstring sports injury prevention

A meta-analysis of the prevention of hamstring injuries was performed on 13 included papers, with a total of 6797 subjects (Figure 5) [4,5,27,39,41,46,47,48,49,50,51,52,53]. The interventions included additional hamstring eccentric training and FIFA11+. The results showed moderate heterogeneity in the effect sizes of the different hamstring eccentric training protocols (I^2^ = 52%, with an RR value of 0.54 and statistically significant at *p* < 0.001).

2.Secondary analyses of the effect of hip and groin sports injury prevention

Ten RCTs with 10,416 subjects were included. The number of reported hip and groin injuries was 122 in the intervention group and 187 in the control group [4,5,27,39,40,42,44,46,54]. The analysis showed low heterogeneity of 38% in the included literature; therefore, a fixed-effects model was used for the analysis. The results of the meta-analysis showed a *p* value of 0.005 and an RR value of 0.73, indicating that hamstring eccentric training, when added to the training program, was effective in preventing hip injuries and groin injuries (Figure 5).

3.Secondary analyses of the effect of knee sports injury prevention

A total of 13,809 knee injury subjects from 17 RCT studies were included in the meta-analysis, with interventions, including hamstring eccentric training in warm-up or injury prevention programs, versus FIFA11+ and the results (Figure 6) showed an RR value of 0.66 for the effect size of the different hamstring eccentric training protocols, with moderate heterogeneity between studies (I^2^ = 54% and statistically significant (*p* < 0.001)) [4,5,18,19,22,27,39,41,42,43,44,45,46,47,55,56,57].

4.Secondary analyses of the preventive effect of ankle sports injuries

Fifteen papers with a total of 14,273 subjects were included, and the results of the analysis (Figure 6) showed low heterogeneity between the papers, with I^2^ = 47%, using a fixed-effects model [4,5,18,19,22,27,39,40,41,44,45,46,47,55,58]. Thus, programs that included hamstring eccentric training were effective in preventing ankle injuries in the athletic population (RR = 0.78). The results of the analysis were statistically significant (*p* = 0.006).

#### 3.5.3. Results of Subgroup Analysis (Table 1)

Exercise frequency: The subgroup analysis was performed on the 13 RCT studies included in the literature, with a total of 10,989 subjects. The results showed low heterogeneity between training once a week, training twice a week (I^2^ = 0%), and high heterogeneity in the amount of effect between groups performing training more than twice a week (I^2^ = 89%). Hamstring eccentric training once a week had no effect on lower extremity injury prevention (RR = 1.00), while training twice a week and training more than twice a week had significant effect sizes. An exercise frequency of twice a week significantly reduced lower extremity injuries (RR = 0.60; 95% CI: 0.49–0.72).

Exercise period: A subgroup analysis was performed on 12 included RCT studies, with a total of 14,712 subjects. The results showed moderate heterogeneity (I^2^ = 67%) between training groups of 10–20 weeks and moderateto high heterogeneity between 21–30 weeks and >30 weeks. Analysis of effect sizes indicated that the most significant effect size was found for training at 21–30 weeks (R = 0.62; 95% CI: 0.38–1.01), followed by the second most effective intervention at 10–20 weeks (RR = 0.68; 95% CI: 0.54–0.87), and a less significant intervention at greater than 31 weeks (RR = 0.77; 95% CI: 0.60–1.00).

Intervention methods: A subgroup analysis was performed on 13 included RCT studies, with a total of 1472 subjects. Results suggested low heterogeneity between the intervention modality group with FIFA11 and in the inclusion of additional Nordic drop centrifugation training (I^2^ = 0% and I^2^ = 37%, respectively) and high heterogeneity in effect size between the group with FIFA11+ intervention (I^2^ = 88%). Effect size analysis showed that the cumulative effect size was more pronounced in the FIFA11+ intervention group (RR = 0.66; 95% CI: 0.48–0.90) and that the FIFA11 intervention modality was the least effective in preventing lower extremity injuries (RR = 0.94; 95% CI 0.85–1.05). The FIFA11+kids intervention was the most effective (RR = 0.52; 95% CI: 0.33–0.83), but there was only one piece of literature in this group and the results were not representative.

Competition level: A subgroup analysis was performed on 15 RCTs with a total of 14,721 subjects. The results indicated low heterogeneity (I^2^ = 0%) in the amateur group and high heterogeneity within the elite and adolescent groups for the hamstring eccentric training protocol. The results of the effect size analysis showed a significant effect of the hamstring eccentric training program on injury prevention in the amateur sports population (RR = 0.54; 95% CI: 0.38–0.75) and a positive effect on lower extremity injury prevention in adolescent and elite athletes.

**Table 1 ijerph-20-02057-t001:** Results of subgroup analysis.

Subgroup	Heterogeneity Test I²(﹪) P-Value		Group	RR (95%CI)	The Number of Study	Sample Size
Exercise methods	37%	0.19	Additional hamstring eccentric training	0.71 (0.59–0.87)	4	4376
	88%	<0.001	FIFA 11+	0.66 (0.48–0.90)	8	6835
	0%	0.82	FIFA11	0.94 (0.85–1.05)	2	2548
	-	-	FIFA11+Kids	0.52 (0.33–0.83)	1	962
Exercise frequency per week	0%	0.38	Once per week	1.00 (0.84–1.18)	2	2357
	0%	0.52	Twice per week	0.60 (0.49–0.72)	5	3329
	89%	<0.01	More than twice a week	0.69 (0.53–0.90)	6	5303
Exercise period	67%	0.03	10–20weeks	0.68 (0.54–0.87)	4	2243
	86%	<0.01	21–30weeks	0.62 (0.38–1.01)	4	3334
	79%	<0.01	More than 31weeks	0.77 (0.60–1.00)	7	9144
Competition level	84%	<0.01	Youth	0.75 (0.56–0.99)	7	9688
	80%	<0.02	Elite	0.73 (0.57–0.93)	6	4762
	0%	0.41	Amateur	0.54 (0.38–0.75)	2	271

“-“ Indicates no data.

## 4. Discussion

The main findings of this study, based on the inclusion of 23 randomized controlled trials, showed that a hamstring eccentric training program adopted in youth and adult sports had a significant positive effect on the prevention of lower extremity injuries (RR 0.72; 95% CI 0.60–0.88) and was effective in reducing hip, hamstring, knee, and ankle injuries. The most significant effect on hamstring injury prevention (RR = 0.54) reduced the risk of injury by 46%, followed by knee injury prevention, with the weakest effect on ankle injury prevention, complementing the results of previous meta-analyses examining lower extremity injury prevention [33,34,59]. Consistent recommendations in high-quality RCTs suggest that the implementation of hamstring eccentric training warm-up exercises can significantly reduce the occurrence of lower extremity sports injuries [4,19,59].

FIFA11+ is proven to be an effective warm-up workout to prevent sports injuries. A meta-analysis showed that FIFA11+ exercises reduced lower extremity sports injuries by 39% [34]. In comparison with previous meta-analyses, we included more high-quality RCTs in the study to expand the sample size. Competition levels, training cycles, and training frequency were also analyzed and discussed to quantify the effect of lower extremity injury prevention. The study showed that FIFA11+ exercises with a systematic process had a positive effect in preventing lower extremity sports injuries, as compared with incorporating additional hamstring eccentric training, which decreased the incidence of lower extremity injuries by 34%.

In training programs, training frequency is an essential factor that affects the effectiveness of the prevention of sports injuries. This study suggested that the greatest effect size (RR = 0.60) for prevention of sports injuries was a twice-a-week hamstring eccentric training program, which is consistent with the study by Lopes et al. [44]. An RCT conducted by Whalan et al. showed that training twice a week in a FIFA11+ warm-up program reduced injuries by 65% in an athletic population [60]. Training frequency above twice a week was second most effective in preventing lower extremity injuries (RR = 0.69); however, eccentric training programs performed only once a week were ineffective in preventing lower extremity sports injuries. Only two papers on exercise frequency were included in this study; therefore, more research on exercise frequency is needed in the future [42,43].

Exercise period also influenced the relationship between training regimen and lower extremity sports injury prevention. Fifteen RCT studies were included, all of which had a training frequency of 1–5 sessions per week, and a subgroup analysis showed that a hamstring eccentric training regimen lasting 21–30 weeks was the most effective in preventing lower extremity injuries, resulting in a 38% reduction in lower extremity injuries (RR = 0.62). This may be due to the fact that the hamstring eccentric training protocol of 21–30 weeks increased the eccentric strength of the hamstrings, improved the muscle balance between the hamstrings and quadriceps, and maximized the promotion of adaptive changes in hamstring strength and structure [23,61,62,63].

We found that the inclusion of hamstring eccentric training was able to prevent lower extremity injuries at all levels of sport, with the implementation of hamstring eccentric training in youth sports being the least effective in preventing lower extremity injuries (RR = 0.75), which was followed by elite athletes in which it was the second most effective (RR = 0.73); this was followed by amateur levels of sports, wherein it was the most effective in preventing sports injuries, reducing lower extremity injuries by 46% (RR = 54%). Decreased heterogeneity between the studies was found by grouping the exercise levels, suggesting that different exercise levels of the population may be the source of heterogeneity.

The mechanism of lower extremity injury can be explained by abnormal lower extremity mechanics, greater knee valgus angle, and torque. Hamstrings play an important role in running exercises, especially at the end of the swing when there is a strong active lengthening muscle action to slow hip flexion and knee extension. All of the warm-up exercises included in the 23 publications included hamstring eccentric training, which can increase peak hamstring centrifugation, increase hamstring activation, and improve the muscle balance between hamstrings and quadriceps, which may reduce the risk of lower extremity injury [64,65].

Biceps femoris long head (BFLH) after NHE produces structural adaptations including increases in length and thickness, while improving the biceps femoris pinna angle [65], which has practical implications for sports injury prevention. Butler et al. evaluated dynamic balance tests on 59 soccer players and showed that increasing the hamstring eccentric training during warm-up exercises may improve dynamic balance function, in increasing neuromuscular control, a key factor in injury prevention and increased athletic performance [33]. Previous systematic evaluations and meta-analyses aimed at reducing sports injury interventions included populations of only soccer players [30,33,34]. A recent meta-analysis included 15 studies evaluating the effect of incorporating NHE into injury prevention programs versus controls for hamstring injuries that reported incidence in women and men across sports and age groups [32]. However, the population of this study was limited to athletes, there was no meta-analysis of injuries in the general sports population and adolescents, and the outcome indicators from previous studies [28,32,35] reported only on the prevention of hamstring injuries and not on the effects of interventions for lower extremity injuries and hip, knee, and ankle injuries.

The meta-analysis in this study validates that hamstring eccentric training during exercise is effective in preventing hamstring injuries and has positive implications for the prevention of lower extremity injuries; therefore, hamstring eccentric training should be used further in future sports injury protection programs and warm-up exercises.

Our research has some limitations. The literature search and selection approach might have produced certain biases. We conducted the meta-analyses only on published RCT-type literature, which thus potentially exposed the study to publication bias. There was high heterogeneity in this study and the results of the subgroup analysis were affected by this. The diversity of hamstring eccentric training protocols and the inclusion of populations with different exercise levels and inconsistent training frequency and training period may have led to a high degree of heterogeneity between the studies. In the implementation of training, it is difficult to blind subjects and coaches, which affects the methodological quality of the meta-analysis. Future studies should therefore refine the experimental design to investigate, in depth, the physiological mechanisms of hamstring eccentric training for sports injury prevention, under the premise of controlling load variables, and to establish more precise exercise prescriptions for different age groups (children, adolescents, and youth) and in different sports domains.

## 5. Conclusions

Hamstring eccentric training has a positive effect on the prevention of lower extremity sports injuries. It is also effective in reducing hamstring, knee, hip, and ankle injuries. A hamstring eccentric training program with an exercise frequency of twice a week and an exercise period of 21–30 weeks is the most effective in preventing lower extremity sports injuries.

## Figures and Tables

**Figure 1 ijerph-20-02057-f001:**
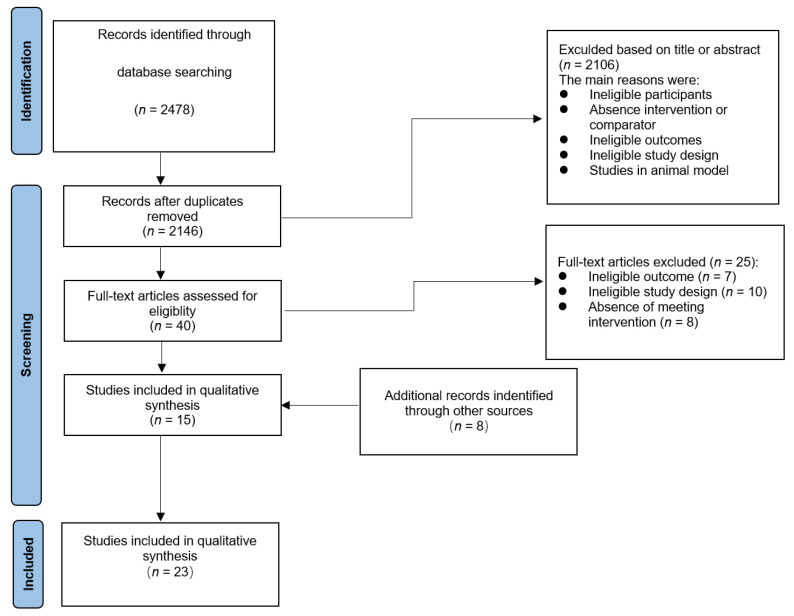
Flow chart of study selection for the analysis of the effects of different hamstring eccentric exercise programs on preventing lower extremity injuries and possible adverse events related to these programs.

**Figure 2 ijerph-20-02057-f002:**
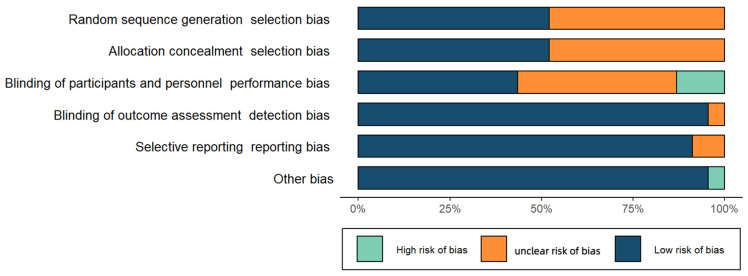
Risk of bias summary. The authors’ judgments about each risk of bias item is presented as percentages across all included studies.

**Figure 3 ijerph-20-02057-f003:**
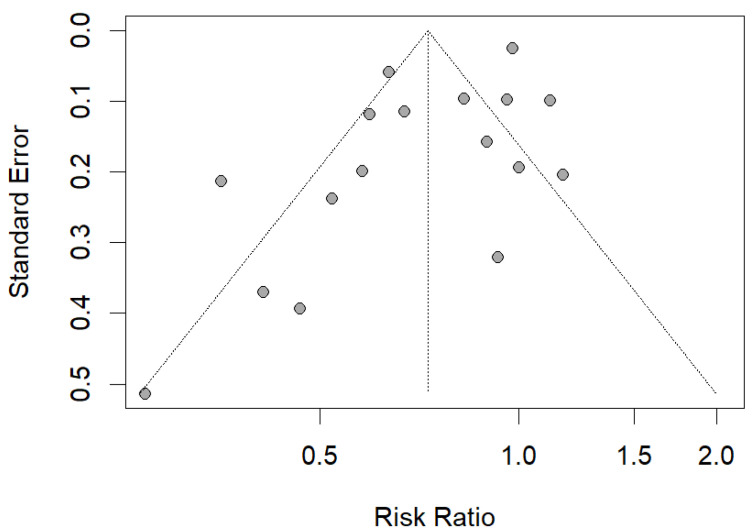
Funnel plot based on study standard error and risk ratio in assessing publication bias.

**Figure 4 ijerph-20-02057-f004:**
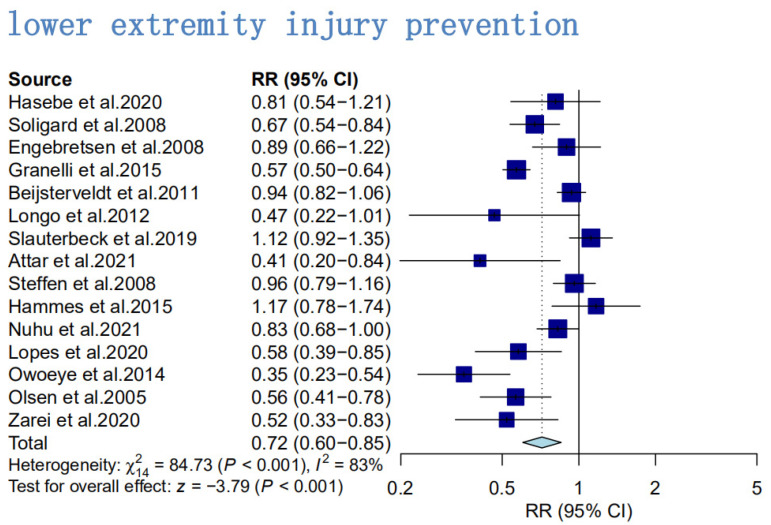
Primary analysis of overall lower extremity injury rates in hamstring eccentric exercise prevention programs compared with control intervention [4,5,18,19,22,27,39,40,41,42,43,44,45,46,47].

**Figure 5 ijerph-20-02057-f005:**
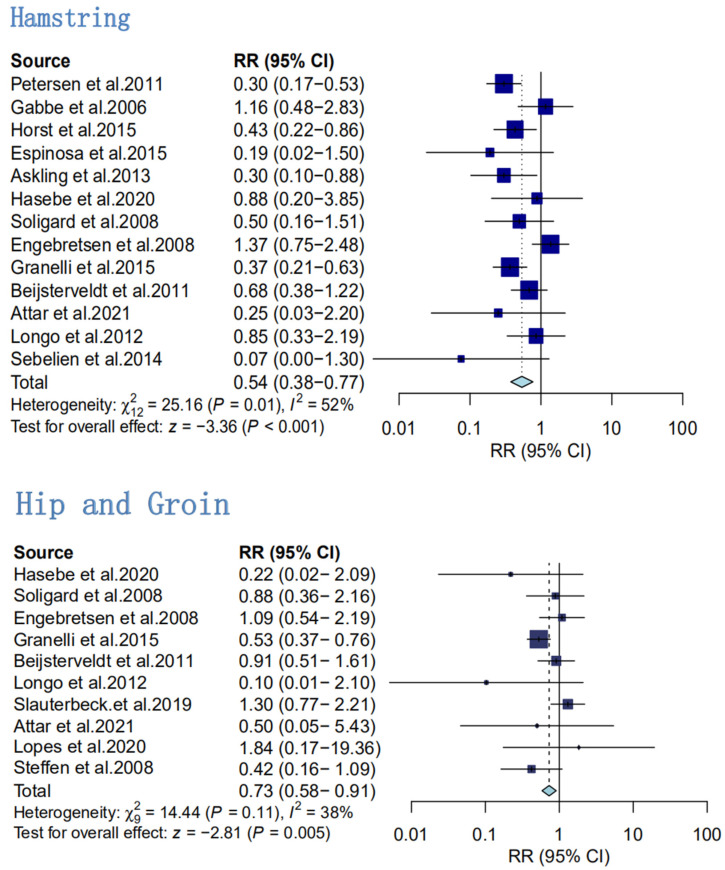
Secondary analyses of injury rates in hamstring eccentric exercise prevention programs, compared with control intervention for the following specific body regions: hamstring and hip and groin [4,5,18,19,22,27,39,40,41,42,43,44,45,46,47].

**Figure 6 ijerph-20-02057-f006:**
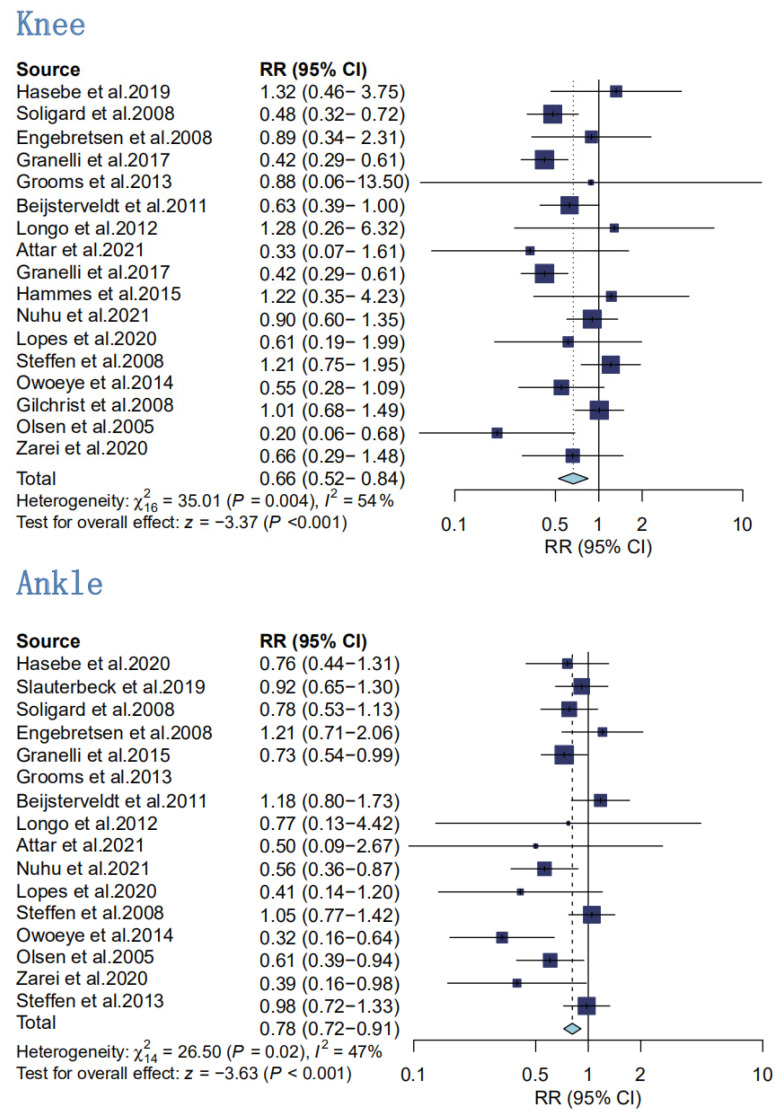
Secondary analyses of injury rates in hamstring eccentric exercise prevention programs compared with control intervention for the following specific body regions: knee and ankle [4,5,18,19,22,27,39,40,41,42,43,44,45,46,47,55,56,57,58].

## Data Availability

The original contributions presented in the study are included in the article/Appendix A. Further inquiries can be directed to the corresponding author.

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
