# Peer review of "Effects of Different Hamstring Eccentric Exercise Programs on Preventing Lower Extremity Injuries: A Systematic Review and Meta-Analysis"

_ijerph, 2023, doi:10.3390/ijerph20032057_

Round 1

Reviewer 1 Report

Dear Authors,

Thank you for the submission of this manuscript, it was a very interesting read, which addresses a contemporary and increasingly important. After careful consideration I did not feel that the paper, as it stands, was acceptable for publication, but with refinement, may add to the body of knowledge.

I have attached a document with more detailed notes that articulates these concerns more deeply. However, I feel that the title, scope of the search strategy and conclusions are misaligned. Essentially, this paper reviews warm-up activities that include one eccentrically focused  activity, of 11 others and then attempts to draw conclusions on the profilactic nature of this one. In fact the injury reduction can not be isolated to one exercise, when it is part of a larger program. 

I encourage the authors to return to this area from a fresh perspective that is considerate of the comments made in the review. 

Author Response

Hello, dear editor

Thank you for your comments, I think they are very interesting and I have made the following additions and changes.

  1. I have added more detailed search strategies in the annexes, so that these search formats can be used to obtain the included literature
  2. After careful consideration, I have revised the conclusion of the article so that it corresponds closely to the title, and have revised the conclusion to a recommendation.
  3. It is true, as you say, that the reduction of sports injuries cannot be isolated to a single training movement should be a systematic training process. However, a large number of randomized controlled trials have shown that the inclusion of hamstring eccentric training is effective in reducing lower extremity injuries compared to the absence of hamstring eccentric training, and this was illustrated in the 23 randomized controlled trials we included. Previous meta-analyses have shown that hamstring eccentric training does enhance eccentric strength and improve muscle balance to prevent sports injuries[1]; the inclusion of hamstring eccentric training prevents lower extremity injuries[2-4].
  4. Compared with the previous meta-analysis, we included more high-quality randomized controlled trials and precisely analyzed the effect of injury in various locations of the lower extremity. In our subgroup analysis, we analyzed and discussed different populations, training cycles, training frequencies and training protocols to improve the previous findings.

Reference

  1. Cuthbert, M.; Ripley, N.; McMahon, J. J.; Evans, M.; Haff, G. G.; Comfort, P., The Effect of Nordic Hamstring Exercise Intervention Volume on Eccentric Strength and Muscle Architecture Adaptations: A Systematic Review and Meta-analyses. Sports medicine (Auckland, N.Z.) 2020, 50, (1), 83-99.
  2. van Dyk, N.; Behan, F. P.; Whiteley, R., Including the Nordic hamstring exercise in injury prevention programmes halves the rate of hamstring injuries: a systematic review and meta-analysis of 8459 athletes. British journal of sports medicine 2019, 53, (21), 1362-1370.
  3. Al Attar, W. S. A.; Soomro, N.; Pappas, E.; Sinclair, P. J.; Sanders, R. H., How Effective are F-MARC Injury Prevention Programs for Soccer Players? A Systematic Review and Meta-Analysis. SPORTS MEDICINE 2016, 46, (2), 205-217.
  4. Gomes Neto, M.; Conceicao, C. S.; de Lima Brasileiro, A. J. A.; de Sousa, C. S.; Carvalho, V. O.; de Jesus, F. L. A., Effects of the FIFA 11 training program on injury prevention and performance in football players: a systematic review and meta-analysis. Clin Rehabil 2017, 31, (5), 651-659.

Reviewer 2 Report

Dear Authors,

I was pleased to review the paper entitled “Effects of Different Hamstring Eccentric Exercise Programs on Preventing Lower Extremity Injuries: a Systematic Review and Meta-Analysis” Int J of Environmental Research and Public Health 

The present paper is very interesting, it focuses on utility of different hamstring eccentric training protocols for prevention of lower limb injuries.

Therefore, it is my opinion that the content is original, current, and relevant.

However, there are some remarks

-       the introduction section is unnecessary lengthy; literature citations should be reduced giving way to a more accurate description of anatomic and physiological function of hamstrings; 

Actual hamstring exercise role in Literature should be better summarized. 

I also suggest the authors to promote  focusing of the aim of the study by introduce open questions about this topic.

-       the discussion section should be better focused on the evidence synthesis from the present study, properly limiting the discussion of data from the literature;

-       the Authors should finally describe a recommended 

-       Novelties and differences if compared to other systematic review already published should be better analyzed

-       Conclusion should better describe desirable upcoming study design able to clear the best protocol of hamstring exercise 

-       please correct line 391 (Another meta-analysis showed51 that FIIFA11 significantly improves dynamic…)

Best regards

Author Response

Dear Editor
I am grateful that you were able to provide me with valuable suggestions for this article, and I have revised the article as a whole based on the suggestions you made
1. the introduction section has been streamlined to describe the hamstring anatomy and physiological function and to summarize the value of hamstring exercise
2. reduced the discussion of data in the discussion section, focusing on the synthesis of evidence and adding innovative points and recent findings to the article
3. After careful consideration, the conclusions have been revised to correspond to the title and training recommendations are provided at the end.
4. All other issues have been revised.
Best regards

Reviewer 3 Report

Dear Authors,

I would like to express my gratitude regarding the opportunity to review this manuscript.

The manuscript at this stage requires considerable improvements. Below suggestions with line indication:

12 – Please insert space “.A”.

17 – Please consider rephrasing and English improvement.

17,18 – Numbers in full and numerical, please standardize the format.

18 – Please consider the numbers with “.” instead of “,”.

31 – Citation numbers should be presented without spaces. This should be considered in all manuscript, for example also in lines 344, 346.

34 – “sports[3].” – Please correct in this line and throughout the manuscript, many other situations.

65 – “Fe´de´ration” – Please correct.

114 – After “:” lowercase, in the same page, above, the criteria was different. Standardization is suggested.

129-131 – Please correct.

130 – The abstract indicated 2022 (also line 142) and in this line 2021. Please clarify.

144 – The start of paragraph format is different compared to others. Please review.

144 – I believe it is “were” instead of “are”. Please confirm.

157-161 – Please revise the text.

186 and 187 – Please review end point spaces.

189 – “Publication bias was examined using funnel plot was used to test.” Please revise the English. The English should be improved in all manuscript.

172-200 – Please consider including references to support the procedures.

205 – Please consider “2.478”.

212 – Please improve the figure quality. Please consider “.” In units. Please standardize the space before and after “=”.

216 – Please insert space after end point.

218-230 – Please standardize numerical or with text.

225 – Please describe “ACL” in full.

230 – Please insert space before parenthesis.

235 – Please improve figure 2 quality.

240 – Please remove the line before 240.

242 – End point is missing.

244 - Please improve figure 3 quality.

247 and 248 – “.” is suggested instead of “,” in the values.

256 – Please improve figure 4 quality. It is also suggested to add to the references the correspondent number, for example “Hasebe et al 2020 [4]”. Please confirm the year of publication, the figure presents 2019 and the reference 2020. The same should be considered regarding figures 5, 6, 7 and 8. Please also revise the text and format of the titles.

263 – Please consider “.” in the value “6.797”. It is suggested to assume this format in all values presented in the manuscript (please carefully review the entire document).

313 – “),and” – Please correct.

332 – Please format the table and footnote considering the journal template and the instructions for authors.

335 – Please remove the space.

367 – Please correct the citation format.

368-369 – “Saleh A et al.” is not ref 38 (and only the last name should be presented in the text, removing the “A”. Please carefully revise all references in the manuscript.

402 – Only “NHE” here, previously abbreviated.

431-435 – Please indicate the study limitations.

436 - It is suggested to reformulate the conclusions section considering clear/direct and take-home messages, if possible, with practical application.

439 – “FIFA”.

446-458 – Please insert end points. Please insert space in 454.

459 – All refs format should be reviewed in detail, considering the journal´s template and instructions for authors.

Please carefully review the English throughout the manuscript.

Author Response

Dear Reviewer:

On behalf of my co-authors, we thank you very much for giving us an opportunity to revise our manu, we appreciate you very much for your positive and constructive comments and suggestions on our manu entitled “Effects of Different Hamstring Eccentric Exercise Programs on Preventing Lower Extremity Injuries: A Systematic Review and Meta-Analysis”. We also sought professionals to systematically revise the English of the article and adjusted the quality and formatting of all images according to your comments.

We would like to express our great appreciation to you and editor for comments on our paper. Looking forward to hearing from you.Thank you and best regards.

Yours sincerely,

Chenxi Hu

Name: Yafeng Song E-mail: songyafeng@bsu.edu.cn

Reviewer 4 Report

Thank you for the opportunity to review this article. The paper addresses a novel under-researched area, which has the potential to provide useful recommendations for coaches. However, there are some questions that need to be addressed to the manuscript.

Specific comments are provided below:

KEYWORDS

Change the keywords that match in the title (line 26)

INTRODUCTION

This section needs to be improved. 

Explain in more depth each paragraph. 

Paragraph 1: Explain in more detail injuries in sports

Paragraph 2: Include the explanation about FIFA 11+ in this paragraph.

Paragraph 3: Explain in more detail hamstring injuries.

Add age of subjects (line 40)

Correct the word federation (line 53)

MATERIAL AND METHODS

It is necessary to update the date (Line 109)

Please include a table with characteristics of study participants and training programs. (line 192)

Do you think these studies are of a high-quality level to be included in the meta-analysis? (line 197)

Capital letter “the”. (line 248)

p value always in italics. Review the entire document.  (line 248)

There are many figures, could you include two, three or four figures in one figure?

Change “mothods” to “methods” (line 289)

Add “.” after “sport” (line 356)

Change “Eccentrictraining” to “Eccentric training” (line 356)

CONCLUSIONS

Include examples of hamstring exercises that might be of interest to coaches and trainers.

Change “FAFA11+” to “FIFA11+” (line 402)

Author Response

(The authors gave the same response as above.)

Round 2

Reviewer 3 Report

Dear Authors,

Thank you for considering my suggestions and incorporating them into the manuscript, which globally improved, congratulations. 

Below small suggestions related to this last version (V4), with line indication.

86 – Please insert the citation number.

212 – I believe “bias” should be in uppercase, in line with other subtitles. Please review throughout the manuscript.

236 – Please carefully revise the figure content. For example, the “=” should always present the same format.

246-248 – Please standardize numbers format.

263-264 / 270-271 / 284-285 – Please revise the space/text format considering the journal template and the instructions for authors.

271,272 – Please review the numbers format “,” and “.”.

280 – In the figure, please place “.” after “et al”. Please consider this in all figures.

 293 – I think this figure was not correct. Please make sure all figures were corrected and always “et al” with “.”.

353 – Please revise in the number “,” or “.”. This should be considered in all manuscript.

498-500 – Please correct the authors contributions considering the journal template and the instructions for authors.

719 – Please correct the references considering the journal template and the instructions for authors. Some examples: Journals should be abbreviated, DOI´s indicated, upper and lowercase in titles should be standardized (for example ref 40 different compared to others), format of ref 60 is not correct. Please also revise the size of letter and paragraph.

A careful final reading after considering the above indications is suggested before V5 finalized.

Author Response

Dear Reviewer:

On behalf of my co-authors, we thank you very much for giving us an opportunity to revise our manu, we appreciate you very much for your positive and constructive comments and suggestions on our manu entitled “Effects of Different Hamstring Eccentric Exercise Programs on Preventing Lower Extremity Injuries: A Systematic Review and Meta-Analysis”.We have detailed explanations and changes in coverletter based on your comments.

We would like to express our great appreciation to you and editor for comments on our paper. Looking forward to hearing from you.Thank you and best regards.

Yours sincerely,

Chenxi Hu

Name: Yafeng Song E-mail: songyafeng@bsu.edu.cn

Reviewer 4 Report

The manuscript has improved notably. Authors have done a big effort in reviewing the manuscript. 

Author Response

Dear Reviewer:

On behalf of my co-authors, we thank you very much for giving us an opportunity to revise our manu.We appreciate you very much for your positive and constructive comments.

We would like to express our great appreciation to you and editor for comments on our paper.Thank you and best regards.

Yours sincerely,

Chenxi Hu

Name: Yafeng Song E-mail: songyafeng@bsu.edu.cn
